

# Variance of vegetation coverage and its sensitivity to climatic factors in the Irtysh River basin

Feifei Han[1,2], Junjie Yan[3] and Hong-bo Ling[1]

[1] State Key Laboratory of Desert and Oasis Ecology, Xinjiang Institute of Ecology and Geography, Chinese Academy of Sciences (CAS), Urumqi, Xinjiang, China
[2] College of Water and Architectural Engineering, Shihezi University, Shihezi, Xinjiang, China
[3] Institute of Resources and Ecology, Yili Normal University, Yining, Xinjiang, China

Corresponding authors
Junjie Yan, yan3550@sina.com
Hong-bo Ling, linghb@ms.xjb.ac.cn

## ABSTRACT

**Background**. Climate change is an important factor driving vegetation changes in arid areas. Identifying the sensitivity of vegetation to climate variability is crucial for developing sustainable ecosystem management strategies. The Irtysh River is located in the westerly partition of China, and its vegetation cover is more sensitive to climate change. However, previous studies rarely studied the changes in the vegetation coverage of the Irtysh River and its sensitivity to climate factors from a spatiotemporal perspective.

**Methods**. We adopted a vegetation sensitivity index based on remote sensing datasets of high temporal resolution to study the sensitivity of vegetation to climatic factors in the Irtysh River basin, then reveal the driving mechanism of vegetation cover change.

**Results**. The results show that 88.09% of vegetated pixels show an increasing trend in vegetation coverage, and the sensitivity of vegetation to climate change presents spatial heterogeneity. Sensitivity of vegetation increases with the increase of coverage. Temperate steppe in the northern mountain and herbaceous swamp and broadleaf forest in the river valley, where the normalized difference vegetation index is the highest, show the strongest sensitivity, while the desert steppe in the northern plain, where the NDVI is the lowest, shows the strongest memory effect (or the strongest resilience). Relatively, the northern part of this area is more affected by a combination of precipitation and temperature, while the southern plains dominated by desert steppe are more sensitive to precipitation. The central river valley dominated by herbaceous swamp is more sensitive to temperature-vegetation dryness index. This study underscores that the sensitivity of vegetation cover to climate change is spatially differentiated at the regional scale.

## INTRODUCTION

Vegetation covers nearly three-quarters of the land surface (*Sarkar & Kafatos, 2004*), which is the main body of the terrestrial ecosystem (*Cramer & Leemans, 1993*), and plays an important role in linking the soil and atmosphere through energy and mass transport

(*Deng et al., 2017*). In the context of global climate change, vegetation is affected by climate change and becomes the sufferers of climate change, acting as an ''indicator'' in global climate change research (*Piao et al., 2006*). The changes vegetation also have feedback effect on climate change (*Liu et al., 2006*; *Sun et al., 2018*). Therefore, monitoring and mapping the sensitivity of vegetation to current climate variability is crucial for projecting future vegetation dynamics and developing sustainable ecosystem management strategies.

Investigating the responses of vegetation to short-term climate anomalies is of great significance to mitigate the ecological, economic, and social consequences of future climate change (*Huete & Alfredo, 2016*). Much current understanding of vegetation's respond to climate change is based on changes in mean climate state (*Mearns, Rosenzweig & Goldberg, 1997*; *Thomas et al., 2004*). We consider this mean state can only represent changes in the equilibrium state (e.g., due to overgrazing or long-term successional cycles (*Clifford, Cobb & Buenemann, 2011*)) instead of anomalies resulting from short-term climate anomalies. However, the response of vegetation to changing climate is comprehensive and often varies dramatically between regions (*Anav & Mariotti, 2011*). Yet, key knowledge of how to identify and then prioritize those regions that are more sensitive to climatic variability is still lacking. However, a key issue must be addressed before we analyze the sensitivity of vegetation to climate variability and that is how to describe vegetation sensitivity in a quantitative way. There are several ways defining the vegetation's sensitivity to climate variability. For example, vegetation's sensitivity refers to the degree and magnitude of vegetation response when the climate anomaly occurs (*You, Meng & Zhu, 2018*) or the degree to which a system changes after a disturbance (*Li et al., 2018*). In this study, vegetation's sensitivity is defined as the magnitude of vegetation response at the moment of the climate anomaly (*Tilman, 1996*). Additionally, *Moulin et al. (1997)* found that due to the relatively slow growth of vegetation, the response of vegetation to climate change often lags, therefore vegetation growth depends both on current disturbances and the residual effects of past climate conditions. We called this phenomenon the vegetation memory effect and it could be described as the persistence of trends in temporal changes of ecosystem properties (*Dash, Carr & Harris, 2014*; *Lhermitte et al., 2010*; *Simoniello et al., 2008*). This 'memory effect' should be considered when assessing the immediate response to short-term climate anomalies (*De Keersmaecker et al., 2015*).

Since the 1970s, the development of satellite remote sensing technology has made it possible for a human to conduct macro dynamic monitoring of earth vegetation from space (*Pettorelli et al., 2014*; *Weng, 2002*). In the past decades, there has been an increase in the availability of satellite data measuring climate and other ecologically relevant variables (*Kerr & Ostrovsky, 2003*). These data offer opportunities to characterize ecosystem sensitivity at high spatial resolution. Based on satellite-derived images, *Seddon et al. (2016)* present recently a novel method to identify ecosystem sensitivity and memory effect to short-term climate variability by developing a vegetation sensitivity index (VSI) that explores the linkage between variability in vegetation productivity (defined as enhanced vegetation index, EVI) and three climate variables (namely air temperature, water availability, and cloud-cover) on monthly time scales. The vegetation sensitivity index is a useful metric to quantitatively assess the sensitivity of different ecosystems to climate variability (*Huete,*

*2016*; *Willis, Jeffers & Tovar, 2018*) and simultaneously takes into account short-term climate effects and vegetation 'memory effect'.

Drylands (including arid and semi-arid regions) occupy over 41% of the global land surface area and are inhabited by >2 billion people (*Mburu, 2017*). Temperature and precipitation are usually the main climatic factors affecting vegetation activities (*Li et al., 2015*; *Seddon et al., 2016*; *Yang et al., 2015*). Arid area is usually characterized by rare precipitation and high temperatures, and both water and heat have important effects on vegetation growth, which makes ecosystems in arid and semi-arid regions are more vulnerable to climatic disturbances (*Rotenberg & Yakir, 2010*). Except for the direct but rare precipitation, soil water supplied by meltwater from ice and snow in mountainous areas is primarily the main water source for vegetation growth in plain areas, especially for forests and wetlands in river valleys. Thus, moisture of the soil is also a profound factor affecting vegetation activities in the arid area. Compared to cloud-cover, soil moisture is far more influential on vegetation in the arid area. TVDI characterizes the changes between vegetation index and land surface temperature, and is an important indicator reflecting the soil moisture status (*Grassini et al., 2010*; *Sandholt, Rasmussen & Andersen, 2002*). So, we substituted the climatic factor of cloud-cover in Seddon's model by TVDI in our study. Additionally, among the all kinds of vegetation indexes, normalized difference vegetation index (NDVI) is the most widely used and also is one of the earliest proposed. No other vegetation index is capable to compare with NDVI in its widely use. NDVI has long been applied in monitoring vegetation dynamics (*Fensholt et al., 2009*; *Slayback et al., 2003*), changes of vegetation phenology (*Jeong et al., 2011*; *Ludeke, Ramge & Kohlmaier, 1996*) and assessing land degradation (*Oba, Post & Stenseth, 2001*; *Thiam, 2003*). NDVI has also been used as basic parameter in modeling of vegetation productivity (*Schloss et al., 1999*), terrestrial evapotranspiration (*Maselli et al., 2014*; *Yang & Wang, 2011*) and predicting yield of crops (*Kastens et al., 2005*; *Mkhabela et al., 2011*). Therefore, we chose NDVI to characterize vegetation coverage. The Irtysh River basin is located in the arid and semi-arid region in the northwest of China where water resources are scarce, and it is an important water source in this area and plays an important role in regional economic development and ecological protection (*Ye & Bai, 2014*). The Irtysh River is an international river, which has complicated interests with neighboring countries in the international distribution of water resources, prevention and control of water pollution, ecological maintenance and international regional cooperation. The existing research (*Huang et al., 2013a*; *Huang et al., 2012*) showed that the Irtysh River basin is sensitive to environmental change. Therefore, we selected this area as our study area and analyzed the changes of its natural vegetation and climate, and further identified the spatial differentiation in the sensitivity of the vegetation to climate changes, which are of valuable reference significance for dealing with climate change and maintaining ecological stability of the basin in the future.

Based on the remote sensing data, we used the Mann-Kendall non-parametric rank statistical test to analyze the vegetation dynamic and then calculated VSI to discuss the vegetation sensitivity. In this study, our aims are to (1) map the spatial trends of vegetation cover in 2000–2018; (2) investigate the controlling factors of vegetation sensitivity and resilience in the Irtysh River basin.

## MATERIALS & METHODS

### Study area

The Irtysh River basin is in the arid and semi-arid area of northwest China where water resources are scare (45°40′∼48°27′N, 85°30′∼91°2′E, Fig. 1). The region relates to the Altai Mountains in the north and crosses into the northern edge of the fold system of the Junggar Basin in the south and is adjacent to Mongolia, Kazakhstan, and other countries in the East and West. A typical temperate continent cold climate (*Ju, Ye & Hu, 2015*; *QiangJi & Wu, 2017*) dominates this area, with annual mean temperature ranging from 3.6 °C to 3.9 °C, and the cold air activities are frequent in winter and spring in mountainous areas, forming disasters such as blizzards, snowstorms, and avalanches now and then. The annual mean precipitation is about 217.1 mm, the precipitation increases gradually alone the increase of elevation, yet the northwest part is wetter than the southeast part in general (*Shen et al., 2007*). The vegetation in the study area is mainly desert meadow and grassland, accounting for 91.58% of the whole vegetated area. The rest are herbaceous swamp distributed in and around the river valley, and broad-leaved forest or shrub scattered in or near the swamp area. As growth of the crops are managed by farmers, which disturbs the regulation of climate, we excluded farm and area of non-vegetation in our study.

### Data source and pre-processing
#### Meteorological data

Monthly precipitation and air temperature datasets of 72 meteorological stations in the territory of Xinjiang province of China, where the study area located, were collected from the China Meteorological Data Service Center (CMDC). The datasets cover the period of 2000–2018. These 72 meteorological stations are unevenly distributed in space, and interpolation is the reliable and common practice to obtain continuous surface data (*Guo et al., 2020*). Both precipitation and air temperature are strongly affected by topography. The interpolation of Australian National University Spline or Anusplin, based on the thin plate spline algorithm, is primarily the first choice when interpolation is necessary, as it takes the effect of topography into account (*Xu & Hutchinson, 2013*). *Hartkamp, White & Hoogenboom (1999)* and *Claire et al. (2001)* have long confirmed the superiority of the thin plate spline over other algorithms, such as inverse distance weighting method or kriging. Datasets of Climatic Research Unit (CRU), WorldClim 1 and WorldClim 2 are all produced using Anusplin, and the China Meteorological Administration also used the Anusplin in producing the daily gridded dataset (*Zhao, Zhu & Xu, 2014*). Therefore, we also applied the Anusplin to interpolate our raster surface of precipitation and air temperature. The pixel size was set to 250 m in the interpolating to keep in line with that of the NDVI images. Among the 72 meteorological stations, 7 are in or at the border of the study area (Fig. 1) (*Jarvis & Stuart, 2001*).

### Remote sensing data

Moderate Resolution Imaging Spectroradiometer (MODIS) NDVI product (MOD13Q1) covering the period of 2000–2018 were used to determine the variation of vegetation cover. This MOD13Q1 product was 16-day NDVI synthetic data using the Maximum

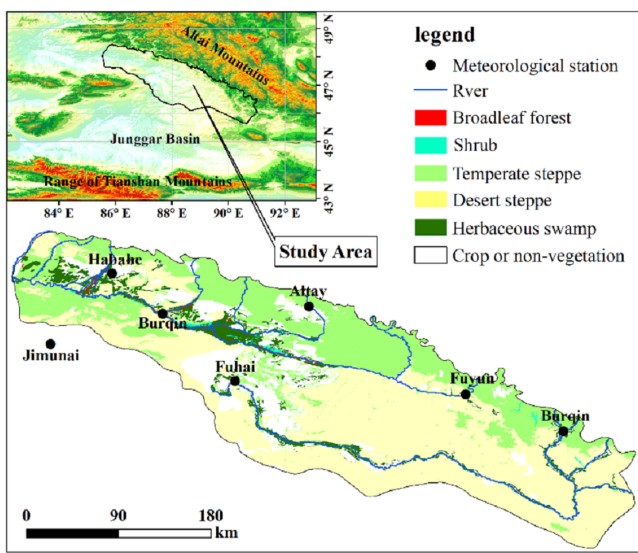

**Figure 1** Overview of the Irtysh River basin.

Value Composite (MVC) method, and its pixel size is about 250 m × 250 m. Time series NDVI images are easily tarnished by noise signal received by the satellite sensors due to effects of the atmosphere, clouds, geometric misregistration or many other uncontrollable factors (*Goward et al., 1991*). In hopes to eliminate the noise and built high-quality NDVI time series, scholars have developed many kinds of smoothing algorithms, such as the SPLINE-curve fitting, double logistic functions, Savitzky-Golay filter, harmonic analysis of time series and so on *Cai et al. (2017)* and *Pan, Hu & Cao (2017)*. Among these smoothing algorithms, both *Pan, Hu & Cao (2017)* and *Cai et al. (2017)* have confirmed the Savitzky-Golay filter's superiority. We followed the recommendation of Pan et al. and Cai et al. and performed the Savitzky-Golay filtering on our NDVI images to obtain high-quality NDVI time series. And then we applied MVC to NDVIs covering every year of the period of 2000–2018 to get NDVI that reflects the yearly highest growing level of vegetation, namely NDVI at yearly time scale. We used yearly maximum NDVI to analyze the inter-annual dynamics of vegetation. We also applied MVC to NDVIs covering every month of the period of 2000–2018 to get the monthly NDVI.

Temperature Vegetation Dryness Index (TVDI) can be used to characterize the degree of soil drought. This study mainly uses the method proposed by *Sandholt, Rasmussen & Andersen (2002)* and *Yao, Zhang & Wang (2004)* to calculate TVDI. TVDI is a combination of vegetation index (VI) and land surface temperature (LST). Monthly TVDI were calculated using the MODIS NDVI and the 8-day composite MODIS temperature product (MOD11A2). The MOD11A2 product includes LST images of day and night, with a pix size about 1,000 m × 1000 m. The day LST was used in the study. Every 4 LST images covering the whole corresponding month were averaged to get the monthly LST time series data, and the pixel size was resampled to 250 m × 250 m with the nearest-neighbor resampling

algorithm (*Christman & Rogan, 2012*; *Khan, Hayes & Cracknell, 1995*) integrated in toolbox of ArcGIS software to match that the of NDVI images.

It is well known that there are ubiquitous data gaps in LST datasets because of non-overlapping satellite orbits, cloud contamination, instrumental malfunction (*Chen et al., 2011*; *Hu et al., 2014*) and interpolation methods are usually applied to fill the data gaps (*Cai et al., 2017*; *Garcia, 2010*). *Garcia (2010)* have developed a fast and robust smooth regression algorithm that combines the Discrete Cosine Transform (DCT) and the Penalized Least Square approach (PLS) together with the Generalized Cross-Validation (GCV) criterion to fill data gaps. *Liu et al. (2020)* tested and applied Garcia's method on the reconstruction of the MODIS LST datasets covering the three continents of South America, Africa and Asia, and confirmed its capability and robustness. We applied this method in our data processing to get a high quality LST dataset and further guarantee our TVDI dataset can reflect the drought of soil more accurately.

## Mann-Kendall non-parametric rank statistical test

When using Mann-Kendall non-parametric test (*Kendall, 1990*; *Mann, 1945*) to test the possible trends of climatic elements and time series, we assume that H0 indicates that the time series $(x1, x2, \ldots, xn)$ are independent of the data sample, and there is no obvious trend; Assuming that H1 is a bilateral test, the distribution of $xi$ and $xj$ are different for all $i, j$ ($i \neq j$), the calculation formula of the statistical variable S of the test is as follows:

$$S = \sum_{i=1}^{n-1} \sum_{k=i+1}^{n} Sgn(x_k - x_i). \tag{1}$$

Among them,

$$Sgn(\theta) = \begin{cases} 1 & \theta > 0 \\ 0 & \theta = 0 \\ -1 & \theta < 0 \end{cases} \tag{2}$$

S is normal distribution, the mean is 0, and the variance is as follows:

$$Var(s) = \left[ n(n-1)(2n+5) - \sum_{t} t(t-1)(2t+5) \right] / 18 \tag{3}$$

where $t$ is the width of each unit. When n>10, $Zc$ converges to a standard normal distribution and can be calculated by the following formula.

$$Z_c = \begin{cases} \dfrac{S_1}{\sqrt{Var(S)}} & S > 0 \\ 0 & S = 0 \\ \dfrac{S+1}{\sqrt{Var(S)}} & S < 0 \end{cases} \tag{4}$$

At a given $\alpha$ confidence level, when $|Zc| > 1.96$, the changing trend reaches a significant level, $|Zc| < 1.96$, the changing trend is not Significant; $Zc > 0$, indicating that the changing trend is increasing, and $Zc < 0$, it is decreasing.

Theil-Sen median trend analysis (*Attaur & Dawood, 2017*; *Coen et al., 2020*) could be used to quantify the trend of time series data, and its calculation formula is as follows:

$$\beta = Median\left( \frac{x_i - x_j}{i - j} \right). \tag{5}$$

In the formula, $1 < j < i < n$, $\beta$ means slope, a positive value means "uptrend", and a negative value means "downtrend".

## Identifying the sensitivity of vegetation to climate variability

The VSI is a novel and empirical metric developed by *Seddon et al. (2016)* that can quantify the sensitivity of different vegetation areas to climate variability (*Huete, 2016*; *Willis, Jeffers & Tovar, 2018*). In this study, we tailored the empirical methodology to identify vegetation sensitive to climate variability on the Irtysh River basin.

Firstly, for the climatic variables, we employed temperature, precipitation, and TVDI, instead of three climate variables as in *Seddon et al. (2016)*. Furthermore, we included the one-month-lagged NDVI monthly data as a fourth variable in the regression to investigate the potential influence of memory effects driving vegetation dynamics.

Secondly, any month with a mean NDVI of <0.1 were excluded to reduce the potential impact of noisy data at low NDVI values, which are attributed to areas with extremely sparse or inexistent vegetation cover (*Zhang, 2015*; *Zhu et al., 2019*). And to remove seasonal component underlying monthly time series, we de-trended the monthly data and then we standardized the de-trended data utilizing the Z-score standardization formula:

$$Z_{i,j} = \frac{x_{i,j} - \overline{x}_j}{\sigma_j} \tag{6}$$

where $x_{i,j}$ is the detrended data in the jth month of the ith year, $x_j$ and $\sigma_j$ are the mean and standard deviation of the variable x in the jth month of all years, respectively.

Thirdly, in this study, the sensitivity of vegetation to climate variability on the Irtysh River basin was primarily calculated using AR1 multiple linear regression approach in each pixel, as follows:

$$NDVI_t = \alpha \times NDVI_{t-1} + \beta \times Tem_t + \gamma \times Pre_t + \delta \times TVDI_t + \varepsilon_t \tag{7}$$

where $NDVI_t$ is the standardized NDVI at time t, $NDVI_{t-1}$ is the standardized NDVI anomaly at time t-1, $Tem_t$, $Pre_t$ and $TVDI_t$ are the standardized temperature, precipitation, and TVDI at time t, respectively. $\varepsilon_t$ is the residual term at time t, and $\alpha, \beta, \gamma, and \delta$ are coefficients for temperature, precipitation, TVDI and $NDVI_{t-1}$ of each pixel, respectively. Each of $\alpha, \beta, \gamma, and \delta$ is a metric of ecosystem stability (Table 1; (*De Keersmaecker et al., 2015*)). Compared to the correlation coefficient which can only indicate whether the ecosystem responds to climate variability, the regressive coefficient can further reflect the response magnitude.

Fourthly, to eliminate the effects of co-linearity between four climate variables, the principal components regression (PCR) was also applied within each pixel to quantify the relative importance of each variable driving variations in the monthly NDVI (*Seddon et al., 2016*). The principal components that had significant relationships with climate ($p < 0.1$) were selected, and we subsequently multiplied the loading scores of each variable by the PCR coefficients. The product scores were summed to estimate the relative importance of each variable in driving monthly changes in NDVI, which provided an empirical approach for mapping the relative importance of climate on vegetation change (climate weights).

**Table 1  Interpretation of the coefficients in the AR1 multiple linear regression approach.**

| Coefficient | Implication | Meaning of absolute value | Meaning of sign |
|---|---|---|---|
| $\alpha$ | Revealing the potential influence of memory effects driving vegetation dynamics. | A large absolute value indicates low resilience, which means that vegetation slowly recovers from previous disturbance. | Positive value of $\delta$ shows NDVI is similar to the previous anomaly. Negative value of $\delta$ shows NDVI is similar to the previous anomaly but with the opposite trend. |
| $\beta/\gamma/\delta$ | Climatic sensitivity index denoting the magnitude of immediate response of vegetation to the contemporary variation in climate variable. | Large absolute values indicate low resistance to temperature/-precipitation/TVDI. | Positive Higher temperature/precipitation/TVDI than average induces a positive NDVI response (higher NDVI). Negative Lower temperature/precipitation/TVDI than average induces a negative NDVI response (lower NDVI). |

The climate weights from each variable were rescaled between 0 and 1 (using the minimum and maximum values of any of the climate coefficient values), to be used for calculations of vegetation sensitivity. To estimate the variations of both the climate variables and NDVI on these time series, we used the residuals of a linear model fitted to the mean–variance relationship of both the NDVI and climate variables for each pixel. We standardized these residuals between 0 and 100 for each variable. Our sensitivity metrics are the log10-transformed ratios of NDVI variability and each of the climate variables. Each ratio was then weighted according to the importance of the climate variable to EVI variability by multiplying it by the value of the regression coefficient (climate weights).

Finally, we summed the sensitivity scores for each of our variables to identify areas of enhanced variability for the period of study.

$$VSI = Tem_{wei} \times Tem_{sens} + Pre_{wei} \times Pre_{sens} + TVDI_{wei} \times TVDI_{sens} \tag{8}$$

where VSI is vegetation sensitivity index, Temwei, Prewei and TVDIwei are the relative importance of temperature, precipitation and TVDI on vegetation change (climate weights), respectively, and Temsens, Presens and TVDIsens are the sensitivity of NDVI to temperature, precipitation and TVDI, respectively. The VSI has no units and therefore provides relative information, wherein a high VSI value is associated with a high response rate of vegetation productivity to climate variability. The detailed algorithm for calculating VSI and the R script can be found in *Seddon et al. (2016)*.

## RESULTS

### Variances in vegetation cover and climatic factors

Based on the annual NDVI data of the study area in 2000–2018, we calculated the annual mean NDVI on a pixel scale and divided it into 6 levels (Fig. 2A) to analyze its spatial pattern. Vegetation coverage in the study area increased from the southern plain to the northern mountain area. Vegetation in the plain area, except for the river valley, had low coverage (NDVI < 0.2) and occupied 68.03% of the vegetated pixels. NDVI of the mountain

areas in the north and some plain areas in the west is about 0.2–0.6, accounting for 24.08% of the vegetated pixels. The central valley area where the wetland and broad-leaved forest are distributed showed high coverage (NDVI > 0.6), accounting for 7.89% of the vegetated pixels.

In the period of 2000–2018, the annually averaged NDVI of the whole study area shows a significant increase trend (statistics $Z_c = 2.17$, $P < 0.05$), the changing rate $\beta$ is 0.0017 (Fig. 3). Spatially, the NDVI of 70.28% of the vegetated pixels showed non-significant increase trend and they are mainly located in the low coverage region dominated by desert meadow and grassland (Fig. 2B). 17.81% of the vegetated pixels showed a significant increasing trend, and are mainly located in the western part and central part in the south. Areas with NDVI showing decreasing trend are sparsely distributed in the northern piedmont area and part of the west end. For the changing rate $\beta$, the proportion of pixels with $\beta>0$ reached 89.04%. NDVI of areas in the central valley, mountains in the east and central plains of the south showed the most rapid increase ($\beta>0.002$), the proportion is 26.79%. The increasing rate was relatively low ($0<\beta<0.001$) for the plain of the east and southwest, and the central mountain of the south, which occupied 62.26% of the vegetated pixels.

Climatic factors are the main driving forces for variation of vegetation. Therefore we performed Mann-Kendall tests on precipitation, temperature and TVDI data to reveal their changing trends separately. The results (Fig. 4) showed that the precipitation and temperature in the study area showed a non-significant increasing trend from 2000 to 2018, meaning whether condition in the study area were getting warmer and wetter. Relatively, TVDI in the study area showed a non-significant decreasing trend from 2000 to 2018, indicating that the soil moisture in the study area is gradually increasing. The increased precipitation and temperature and decreased TVDI indicate that the hydrothermal conditions and soils required for vegetation growth in the study area have been greatly improved in the past 19 years, which promoted the increasing of the NDVI for the whole study area.

## Vegetation memory effects

Vegetation memory effects have been widely reported in water-limited ecosystems at various time-scales (*Los et al., 2006*; *Schwinning et al., 2004*). Seddon et al. found that a one-month lag provided the best explanatory power for vegetation responses to variability on short-term timescales. Therefore, we included the one-month-lagged NDVI monthly data as a fourth variable in the regression to investigate the potential influence of memory effects driving vegetation dynamics. The areas with high variance explained by the t −1 variable in the AR1 model, indicating systems where memory effects play a more important role than contemporary climate conditions in determining vegetation cover (Fig. 5). The larger the t-1 coefficient weight (that is, coefficient $\alpha$), the stronger the memory effect, and the weaker the sensitivity (with lower VSI).

Vegetation showed strong memory effects ($\alpha>0.4$) across almost the whole study area (Fig. 5), especially in some parts of the eastern plain where the coefficient reached more than 0.6, and the area with $\alpha>0.6$ accounted for 27.27% of the vegetated pixels in the

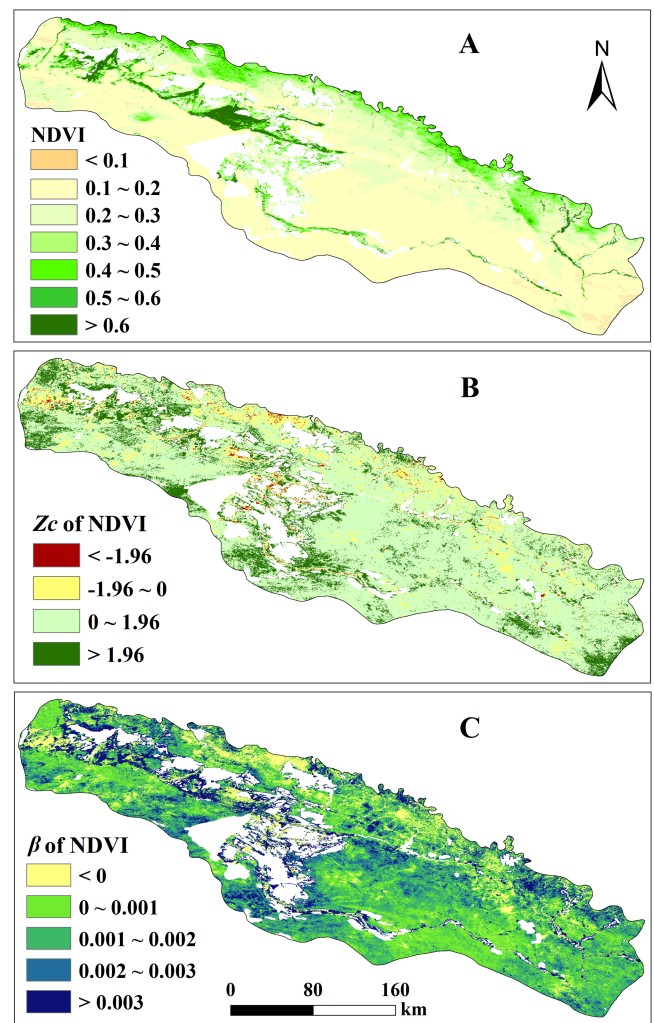

**Figure 2** **Distribution map of annual average NDVI levels, NDVI change trend Zc value and change rate $\beta$ value.** (A) Distribution map of annual average NDVI levels in the study area from 2000 to 2018 and Spatial distribution of variance in vegetation cover; (B) NDVI change trend Zc value and (C) change rate $\beta$ value spatial distribution map. Characterizing change trend of each vegetated pixel in the study area.

study area. Coefficient $\alpha$ of areas in the northwest, the border of the east and river valley in the middle is relatively small ($\alpha<0.4$), indicating weaker memory effects, and the areas proportion is 20.60%. Yet for most parts of the study area, the coefficient $\alpha$ is about 0.4–0.6, and the area proportion reached 52.13%. Notably, vegetation with big NDVI showed weak memory effect in general, such as the herbaceous swamp and broadleaf forest in the river valley and grassland in the mountain area of the north border. In contrast, the desert grass in the plain area showed strong memory effect.

As shown in Fig. 6, the memory effect tends to change along the gradient of NDVI and climatic factors. To identify the controlling factors for vegetation memory effects, we regressed $\alpha$ (t-1 coefficient weight) against three climatic factors (precipitation, temperature, and TVDI) and vegetation cover (defined as NDVI). And considering the
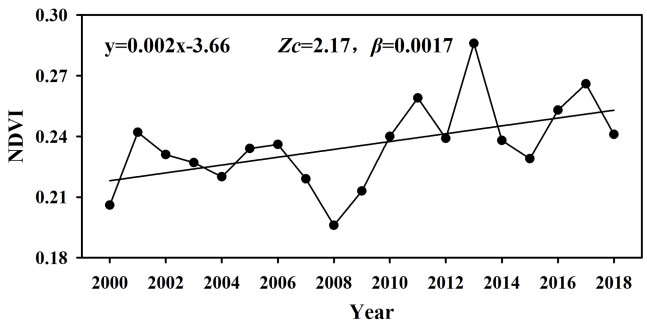

**Figure 3** Interannual variation curve of overall NDVI in the study area from 2000 to 2018.

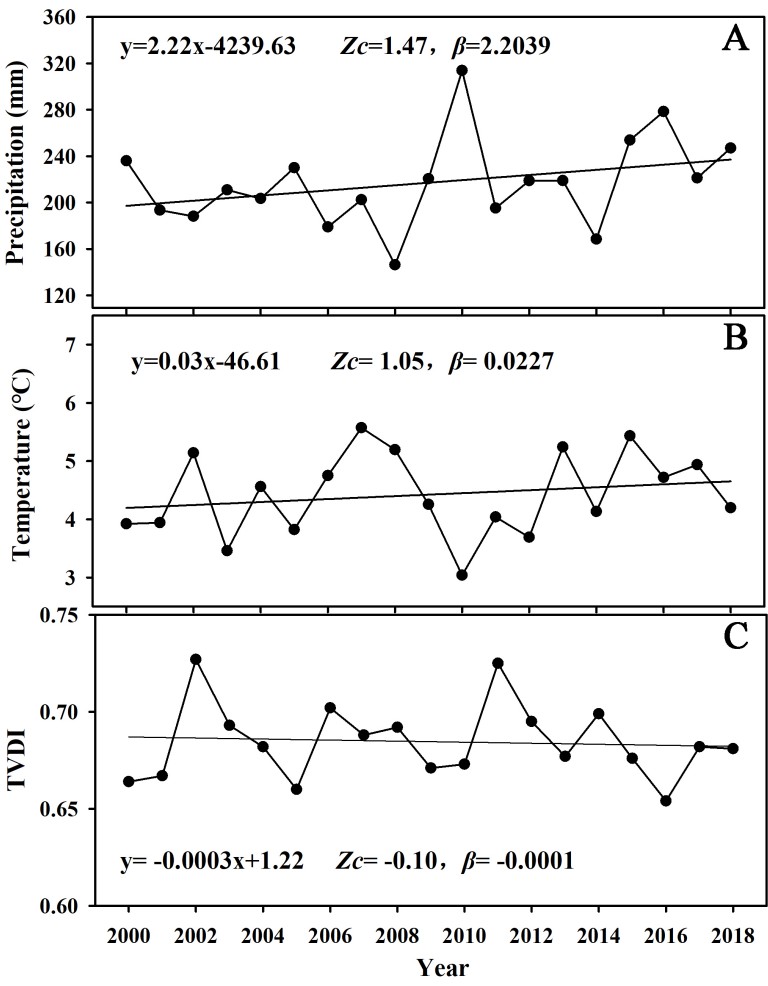

**Figure 4** Interannual variation curves of overall precipitation (A), temperature (B), and TVDI (C) in the study area from the period 2000–2018.

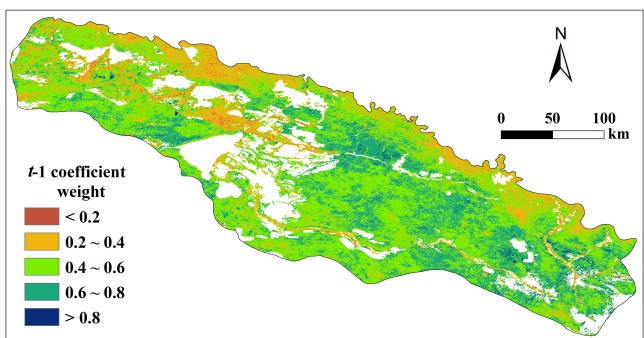

**Figure 5  Spatial distribution of t-1 (AR1) coefficient weight (that is, coefficient $\alpha$) from monthly multiple regression between vegetation cover (defined as NDVI), vegetation cover at t-1, and three climatic variables.** Characterizing the memory effects of vegetation cover in the Irtysh River basin during 2000–2018.

hydrothermal conditions required for vegetation growth, we selected the mean NDVI, mean precipitation, mean temperature, and mean TVDI of the growing season (GS) to participate in the regression. The results showed that vegetation memory effects ($\alpha$) decreased logarithmically as NDVI increased ($R^2 = 0.67$, $P < 0.05$, Fig. 6A). Relatively, vegetation memory effects ($\alpha$) increased logarithmically as TVDI increased ($R^2 = 0.407$, $P < 0.05$, Fig. 6D). Specifically, vegetation memory effects ($\alpha$) presented a quadratic parabola relationship with both precipitation ($R^2 = 0.202$, $P < 0.05$, Fig. 6B) and temperature ($R^2 = 0.155$, $P < 0.05$, Fig. 6C).

## Vegetation sensitivity to climatic variables

Compared to strong memory effects, the VSI in the study area is rather low, with 60.69% of the vegetated pixels where VSI is less than 30 (Fig. 7B). Spatially, areas of big VSI (VSI>30) generally overlap that of weak memory effects ($\alpha$<0.4), such as the grassland in the north border and herbaceous swamp and broadleaf forest in the river valley, indicating that areas with higher NDVI usually shows weaker memory effects and higher sensitivity to climate variability over the past 19 years. Areas of the desert plain show low sensitivity (VSI>30) to climate variability, and overlap the areas of strong memory effects ($\alpha$>0.4).

The relative importance of three climate variables (temperature, precipitation, and TVDI) to vegetation sensitivity also displayed clear spatially heterogeneity across the study area (Fig. 8). Most areas are more sensitive to precipitation, mainly distributed in the southern and central plains dominated by desert meadow. While variation in vegetation cover (defined as NDVI) of the southeastern areas were mainly affected by a combination of precipitation and temperature, and the northern part of this area is affected by a combination of TVDI and temperature. Additionally, vegetation cover in the northwest areas was mainly driven by precipitation and TVDI. Remarkably, the central river valley dominated by herbaceous swamp was more sensitive to TVDI. And the mountain areas with higher elevations in the north are more sensitive to both temperature and precipitation.

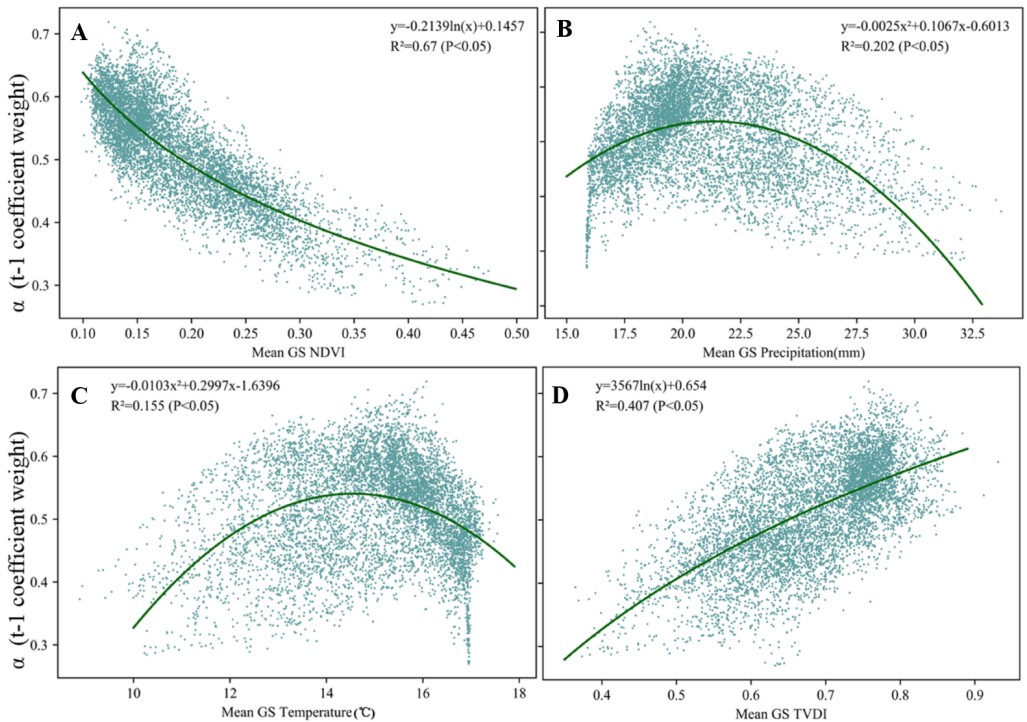

**Figure 6** **Scatter plots of $\alpha$ (t-1 coefficient weight) along mean growing season climatic factors.** Correlations between $\alpha$ (t-1 coefficient weight) and mean growing season (A) NDVI, (B) precipitation, (C) temperature, (D) TVDI in the Irtysh River basin during 2000–2018. The green curves indicate the fitted regression lines.

## DISCUSSION

### Disentangling the driving factors for variations in vegetation cover

The Irtysh River basin is located in arid and semi-arid region. Scarce precipitation and high temperature lead to large evapotranspiration and low soil water storage in this area, which is not conducive to the growth of vegetation, especially in the low land of the southern plain. Therefore, most part of the study area is dominated by desert vegetation. However, as the altitude increases, the precipitation increases and temperature decreases (*Navarro et al., 2020*), and this relieves the severe climatic restrictions. So, grassland in areas of high altitude, mainly the mountain areas in the north, is well developed and the vegetation coverage is also high (0.4<NDVI<0.6). The river valley can rely on the rivers to supply ample water required for vegetation growth, so the herbaceous swamp and broadleaf forest with the highest NDVI (NDVI>0.6) are well developed in this area.

The results of the Mann-Kendall trend test of climatic factors show that the temperature and precipitation in the study area showed an increasing trend from 2000 to 2018, and the TVDI showed a decreasing trend. This is consistent with the findings of *Huang et al. (2013b)*. The analysis of the results indicate that the hydrothermal and soil conditions required for vegetation growth in the Irtysh River basin have been greatly improved. However, with the change of altitude gradient and climatic factors, the variation trend

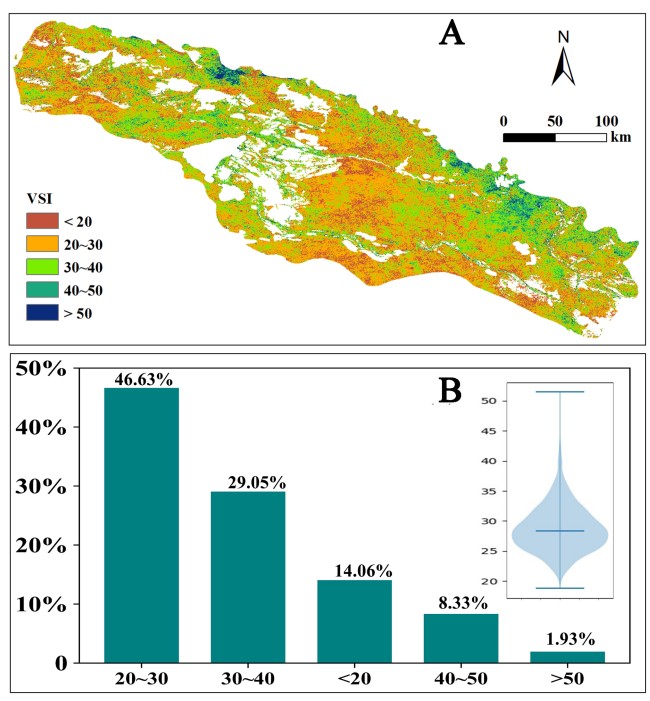

**Figure 7   Distribution of Vegetation sensitivity index (VSI) in the Irtysh River basin during 2000–2018.** (A) Spatial distribution of Vegetation sensitivity index (VSI) in the Irtysh River basin during 2000–2018. The index ranges from 0 (low sensitivity) to 100 (high sensitivity). (B) VSI distribution histogram, insert panel of violin plot shows the frequency distribution of pixel VSI values.

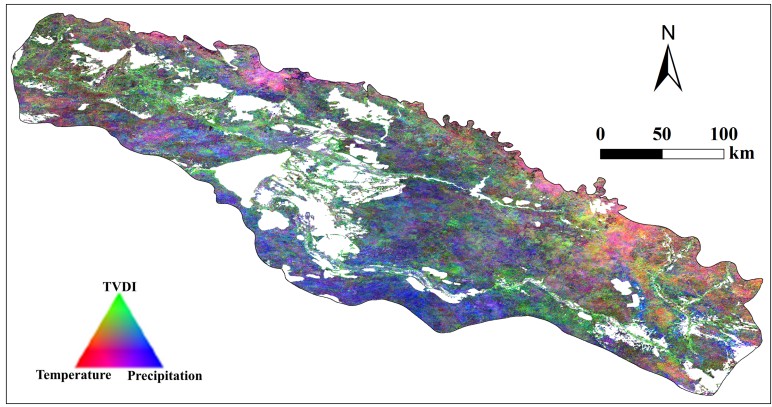

**Figure 8   RGB composite of climate weights from monthly multiple regression between vegetation cover (defined as NDVI), vegetation cover at t-1, and three climatic variables.** Notably, temperature, red; TVDI, green; and precipitation, blue.

(Zc) and rate ($\beta$) of vegetation cover show obvious spatial heterogeneity. The increase of rainfall brings abundant water resources to the desert plain area in the southern study area, improves the available moisture content of the local soil, thus promoting the absorption of plant nutrients, which is conducive to the improvement of water utilization rate of plants,
and greatly promotes the growth of vegetation and the increasing rate of local vegetation cover. The vegetation growth in the central valley, which is dominated by herbaceous marshes, mainly depends on rivers to supply groundwater to meet the requirements of soil moisture. In recent years, with the continuous strengthening of national ecological protection, the ecological water volume of Irtysh River has been well guaranteed (*Yang et al., 2012*; *Ye & Bai, 2014*), which can timely recharge the groundwater aquifer in the valley area, thus increasing the soil moisture in this area, and TVDI presents a decreasing trend. In addition to the supply of rainfall, the vegetation cover in this region showed a significant trend of increase from 2000 to 2018, and the increase rate was also very fast. Studies (*Jiang et al., 2017*) have shown that the impact of air temperature on vegetation growth is topographically different. In the central river valley where it is relatively wet, elevated temperature can promote plant photosynthetic activity and thus lead to a positive response of vegetation growth. In the northern study region where it is relatively dry, increase of temperature can intensify the water deficit through elevated evaporation and thus causes a negative response of NDVI. In particular, although the climatic conditions in the study area showed a pattern of improvement, the impact of human activities on vegetation cover could not be ignored. The research results of Yang et al. (*Han, Xuegang & Yaqi, 2013*) showed that from 1990 to 2010, the overall landscape pattern of the Irtysh River Basin tended to be fragmented, with serious spatial heterogeneity, which was increasingly affected by human activities over time. The research results of this paper also reflect this phenomenon. The regions with different elevations have different responses to the same climatic factors.

## Environmental impacts on vegetation memory effect during growth season

We can see in Fig. 5 that areas with the strongest memory effects are generally located in the desert pain of the southeast, where the NDVI is the smallest and the drought the strongest. And areas with weakest memory effects is gathered in the mountain areas of the north and river valleys in the middle, where the NDVI is the biggest and the drought the weakest. This character in the matching between the memory effects and both of NDVI and drought can also be seen in the clean decreasing trend of metric $\alpha$ along with the increasing NDVI and also the increasing trend with the increasing TVDI (Figs. 6A, 6D). Vegetation in the arid area or desert are usually characterized by their strong capability to coping with disturbances in climatic factors, this can be seen in the constant and largely stable low productivity conditions despite large climate variability and also strong cyclical variability with periods of very low and stable NDVI. So vegetation of these areas usually show strong memory effects. This contrasts to areas with high NDVI, such as the river valley and the mountain area, where the river water and the more precipitation can moderate the severe drought and provide better conditions for vegetation growth, yet the growth of vegetation is restricted by the variation in water supply.

In addition, vegetation memory effects in the study area does not show a clear linear relationship with temperature and precipitation, but we can see from the figure (Figs. 6B, 6C) that there is an obvious inflection point in the image, which means there a threshold in

both the precipitation and temperature effects on memory effects. This might be related to the co-effect of temperature and precipitation on vegetation. For areas in the arid region, altitude usually controls the spatial differentiation of precipitation and temperature, therefore relationship between the memory effects and climatic factors are branded with the influences of altitude on temperature and precipitation. Plain of low altitude is usually characterized by high temperature and scare precipitation, whereas mountain areas are usually characterized by low temperature and abundant precipitation. So, the limiting factor on the growth of vegetation changes gradually from precipitation to temperature along the variation in altitude, which results in vegetation changes in certain areas are co-affected of temperature and precipitation and the threshold in both the precipitation and temperature effects on memory effects.

## Spatial heterogeneity of VSI distribution

VSI reflects the sensitivity of vegetation cover to climate change, and we can identify regions that exhibits amplified responses to climate variability through VSI. While Memory effect measures the capability of vegetation returning to its normal state after suffering the disturbance. Specially, areas with low VSI values showed the largest memory effect (*Seddon et al., 2016*), which is consistent with our study results (Figs. 5 and 7A). The desert plain area in the south of the study area has low sensitivity to climate change and strong vegetation memory effect. When the adverse conditions for vegetation growth are generated due to the vicious climate development or other disturbances in the region, the vegetation will make a hysteresis response to such changes, so that the ecosystem can make timely adjustments to the environmental deterioration. Different types of vegetation respond differently. The main vegetation type is desert meadow, which is a short-lived plant that survives in arid areas by escaping drought (*Guo, Liang & Liu, 2004*; *Lu et al., 2019*). The ephemeral plants germinate and grow quickly in spring when the elevated temperature melts the frozen soil or the covering snow and complete their life cycle before the coming of the hot and dry summer. In addition, studies (*Junzheng & Haojun, 2015*) have shown that the water content of soil at different depths is affected differently by precipitation. Among them, the water content of shallow soil (0–20 cm) is most affected by precipitation. Desert meadow in arid and semi-arid areas have relatively short rooting system that mainly absorb moisture from shallow soil, so the vegetation changes in this area are more sensitive to precipitation.

In contrast to the desert plain area in the south, the central Irtysh River valley showed higher sensitivity and weaker memory effect. The central river valley area is dominated by herbaceous swamps, which are typical low-level swamps, with year-round accumulation of water or drenched soil (*HY et al., 2020*), and the water supply mainly depends on the Irtysh River. The local vegetation is dominated by perennial plants such as caress and gramineous plants. Many plants have short root systems and mainly absorb shallow soil water, which are highly dependent on soil moisture conditions. Therefore, when the amount of river water decreases, the soil water content of the swamp will also decrease, exerting a hard impact on the growth of herbs. Additionally, TVDI is an index reflecting soil moisture, smaller TVDI indicates that the wetter soil (*Sandholt, Rasmussen & Andersen, 2002*), so the vegetation on the site is more sensitive to TVDI.

It is worth noting that the northern mountain regions show strong sensitivity to climate change (VSI>50 in some areas). And the vegetation variation in this area is mainly affected by the combined effects of temperature and precipitation. Compared to the powerful ability of the desert plants in coping the severe drought, temperate steppe in the mountain area is well developed because the elevated altitude relieves both the restrictions of scarce precipitation and high temperature and provides hydrothermal conditions suitable for the grass plants, which makes variations in the coverage of temperate steppe are sensitive to both the precipitation and temperature.

## CONCLUSIONS

This study applied a new method and remote sensing datasets of high temporal resolution to quantify the sensitivity and memory effects of vegetation in the Irtysh River, and further reveal the mechanism of vegetation response to climate change at the regional scale. We found that the variation trend of precipitation, temperature and TVDI all indicated that the climate condition of the Irtysh River basin had been greatly improved, and the vegetation coverage also showed overall increasing trend. From the south to the north of the study area, with the change of topographic and geomorphic features, the memory effect of vegetation and its sensitivity to different climatic factors showed obvious spatial heterogeneity. It is mainly manifested in the following aspects, the memory effect of vegetation in the southern desert plain was stronger and the plants there are more sensitive to precipitation, while the herbaceous swamp and broad-leaf forest in the central valley showed weaker memory effect and were more sensitive to TVDI. The temperate steppe in the northern mountain is highly sensitive to climate change and were more affected by the combination of both precipitation and temperature. These results will help us locate different ecological protection environment types more accurately in the future basin management process, and develop optimal adaptive ecological protection strategies to protect this vulnerable ecosystem.

### Funding

This work was supported by the Xinjiang Tianshan Youth Program (2019Q006), West Light Foundation of Chinese Academy of Sciences (2019-XBQNXZ-A-001) and the Science and Technology Service Network Project of the Chinese Academy of Sciences (KFJ-STSQYZD-114). The funders had no role in study design, data collection and analysis, decision to publish, or preparation of the manuscript.

### Grant Disclosures

The following grant information was disclosed by the authors:
Xinjiang Tianshan Youth Program: 2019Q006.
West Light Foundation of Chinese Academy of Sciences: 2019-XBQNXZ-A-001.
Science and Technology Service Network Project of the Chinese Academy of Sciences: KFJ-STSQYZD-114.

## Competing Interests

The authors declare there are no competing interests.

## Author Contributions

- Feifei Han performed the experiments, analyzed the data, prepared figures and/or tables, and approved the final draft.
- Junjie Yan analyzed the data, prepared figures and/or tables, authored or reviewed drafts of the paper, and approved the final draft.
- Hong-bo Ling conceived and designed the experiments, authored or reviewed drafts of the paper, and approved the final draft.

## Data Availability

Monthly precipitation and air temperature datasets of 72 meteorological stations in the territory of Xinjiang province of China, where the study area located, were collected from the China Meteorological Data Service Center (CMDC). It is available at http://data.cma.cn/en/?r=data/detail{&}dataCode=SURF_CLI_CHN_MUL_DAY_CES_V3.0. The selected time range is January 1, 2001 to December 31, 2018.

Due to copyright restrictions, the CMDC requires registration to access the dataset.

Moderate Resolution Imaging Spectroradiometer (MODIS) NDVI product (MOD13Q1) covering the period of 2000-2018 were used to determine the variation of vegetation cover. It is available at https://ladsweb.modaps.eosdis.nasa.gov/search/order/4/MOD13Q1--6/2000-01-01..2018-12-31/DNB/Country:CHN. The selected product is MOD13Q1 (16-day NDVI synthetic data). The selected time range is January 1, 2001 to December 31, 2018. The selected location is China. It is available at https://e4ftl01.cr.usgs.gov/MOLT/MOD13Q1.061/ and LST datasets can be downloaded at https://e4ftl01.cr.usgs.gov/MOLT/MOD11A2.061/.

## Supplemental Information

Supplemental information for this article can be found online at http://dx.doi.org/10.7717/peerj.11334#supplemental-information.

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
