# Peer review of "Variance of vegetation coverage and its sensitivity to climatic factors in the Irtysh River basin"

_PeerJ, doi:10.7717/peerj.11334_

## Round 0.1 · original submission · Major Revisions

Dear Authors, three reviewers carefully read your manuscript and delivered the same verdict - major revision, which I agree with also. You will find their comments at the bottom of this message as well as two attachments with some details. All this will help you very much to improve the manuscript before re-submission of your paper.

·

Basic reporting

No comment.

Experimental design

No comment.

Validity of the findings

No comment.

Additional comments

Overall, the work is good.Please check the attached comments.

Reviewer 2 ·

Basic reporting

This manuscript adopted a vegetation sensitivity index (VSI) based on high-resolution remote sensing
datasets to study the sensitivity of vegetation to climatic factors in Irtysh River basin. It is a beneficial trial for the vegetation cover monitoring. But the manuscript need major revision before publication.

Experimental design

Figure 1, What is the white regions of the land cover data?

Figure 1, It is better to add the stations' name.

Line 126, How about the interpolation accuracy?

Line 132, Why do you use Savitzky-Golay filter? Add the reference for this method.

Line 140, How to resample?

Line 143, How about the data filling accuracy?

Line 206, What is the

Validity of the findings

Figure 2, line 214, The NDVI value need to be divided into more levels especially in the low values.

Figure 6, The correlation analysis is meaningless for the whole pixels.

·

Basic reporting

1. The introduction is rather vague and needs more detail. I propose that you enhance the description at lines 45-48 to provide more justification for your research gap.
2. In line 52 consider removing “that”.
3. Line 58; “, that is,” to ‘and that is’?
4. Line 64; consider to clarify “in that its state” or alternatively rewrite the sentence.
5. Line 65 to 67; what is the “memory effect”? First define it and then refer to it later on.
6. Line 67 to 69; I suggest sticking to one of the definitions.
7. Line 80; consider to rewrite it.
8. Line 84; rewrite, please.
9. Line 85 to 90; please clarify the point you want to make.
10. Line 95; “At the same time” is not necessary.
11. Line 98 to 101; rewrite it and make your point clear.
12. Figure 1; please explain the reason that one of the meteorological stations are outside of the study area? Maybe you can overlay a hill shade to the zoomed area? The north sign is not necessary as your map has already N-E labels, this also applies to other graphs to avoid redundancy.
13. Line 126; reference is needed here.
14. Line 131; please add a reference.
15. Line 131; please substitute the word “pix” to “pixel” throughout the study.
16. Line 144; make a complete sentence, rewrite.
17. Line 174; provide a reference, please.
18. Line 200; make a complete sentence.

Experimental design

19. Line 132; please explain your choice of smoothing filter and add a reference and also please elaborate the parameters ( you can use THIS LINK as a source; https://www.mdpi.com/2072-4292/9/12/1271/pdf or https://opengeospatialdata.springeropen.com/articles/10.1186/s40965-017-0038-z)
20. Line 143; You can add one or two more sentences here as the gap-filling approach is vital to this study. Here are some links that might be helpful. https://www.mdpi.com/2072-4292/12/3/361/htm and https://www.ncbi.nlm.nih.gov/pmc/articles/PMC4008475/.

Validity of the findings

21. Line 372; Please consider changing “comprehensive” as this study didn’t provide evidence about the comprehension of the method. Alternatively, you can provide evidence that the method is comprehensive e.g., by comparing to similar topics like https://www.mdpi.com/2072-4292/11/21/2475, https://www.mdpi.com/2072-4292/12/19/3150 and or https://www.mdpi.com/2072-4292/12/11/1758, https://www.mdpi.com/2072-4292/12/7/1113.
22. Line 374 to 376; Is this a conclusion of your study, if yes, please explain it?
23. Line 384 to 385; please either bring a sentence about the mentioned analysis or consider removing the sentence.

Additional comments

In general, I like the research the technical part was interesting and all the codes and function were well written and well documented. I encourage the writer(s) to continue writing code as small functions that can be used later easily. I also encourage the writer(s) to be more careful about scientific writing and paying more careful attention to grammar.

---

## Round 0.2 · accepted · Accept

We received two positive reviews on the corrected version of your manuscript after the second round of reviewing. The manuscript has been greatly improved and all reviewers' suggestions have been applied. We believe that the paper is now ready for publication. Please consider removing the abbreviations in the Abstract as they are not needed there but in the body of the text. Probably this can be done by PeerJ staff during technical editing.

·

Basic reporting

no comment

Experimental design

no comment

Validity of the findings

no comment

Additional comments

I have looked through the response and found the authors have successfully addressed my comments. Therefore, I recommend to publish it in the present form.

The manuscript has been greatly improved and all reviewers' suggestions have been applied. I believe that the paper is now ready for publication.

·

Basic reporting

- Please consider removing the abbreviations in the abstract as they are not needed there but in the body text.

Experimental design

No comment.

Validity of the findings

No comment.

Additional comments

No comment.